# Towards Multi-view Consistent Graph Diffusion

## ABSTRACT

Facing the increasing heterogeneity of data in the real world, multi-view learning has become a crucial area of research. Graph Convolutional Networks (GCNs) are powerful for modeling both graph structures and features, making them a focal point in multi-view learning research. However, these methods typically only account for static data dependencies within each view separately when constructing the topology necessary for GCNs, overlooking potential relationships across views in multi-view data. Furthermore, there is a notable absence of theoretical guidance for constructing multi-view data topologies, leading to uncertainty regarding the progression of graph embeddings toward a consistent state. To tackle these challenges, we introduce a framework named energy-constrained multi-view graph diffusion. This approach establishes a mathematical correspondence between multi-view data and GCNs via graph diffusion. It treats multi-view data as a unified entity and devises a feature propagation process with inter-view awareness by accounting for both inter-view and intra-view feature flow across the entire system. Additionally, an energy function is introduced to guide the inter- and intra-view diffusion, ensuring the representations converge towards global consistency. The empirical research on several benchmark datasets substantiates the benefits of the proposed method and demonstrates its significant performance improvement.

## CCS CONCEPTS

• **Information systems** → **Multimedia information systems**; • **Computing methodologies** → *Semi-supervised learning settings*.

## KEYWORDS

Multi-view learning, graph diffusion, graph convolutional networks.

## 1 INTRODUCTION

The growth in multimedia technology has significantly enhanced the capability to gather real-world data from diverse sources, leading to the emergence of multi-view data. This variety of data encapsulates richer information by covering multiple facets of the entities under study. This type of data contains more comprehensive information by encompassing various aspects of the entity under study. In this context, it is crucial to strategically leverage a limited set of labeled samples to infer labels of vast amounts of unlabeled data. Multi-view semi-supervised classification emerges as a pivotal approach to tackle this challenge. Simultaneously, lever-

*ACM MM, 2024, Melbourne, Australia*

© 2024 Copyright held by the owner/author(s). Publication rights licensed to ACM.
ACM ISBN 978-x-xxxx-xxxx-x/YY/MM
https://doi.org/10.1145/nnnnnnn.nnnnnnn

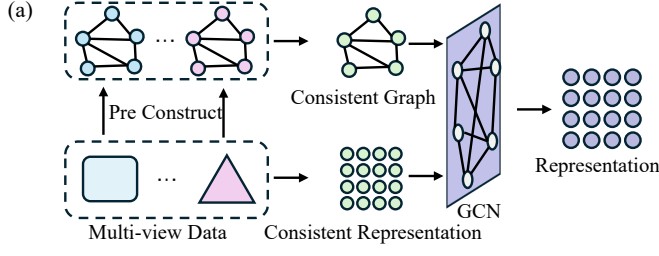

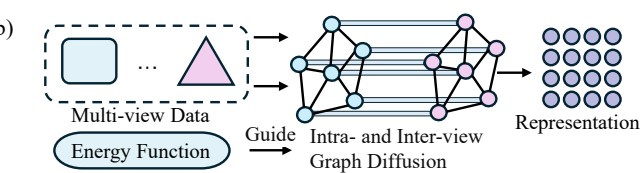

**Figure 1: Comparison of GCNs exploring consistency on multi-view dataset. Subfigure (a) shows previous methods leveraging pre-constructed topologies and shared representations for graph embeddings. Subfigure (b) illustrates the proposed model, which conducts both intra- and inter-view dynamic interactions and incorporates an energy function to guide this process.**

aging graphs to capture complex, irregular structures across diverse fields has garnered significant attention. Graphs excel in depicting intricate relationships, such as those observed among individuals in social networks [27, 43, 44] and the interactive forces between molecules [8, 22, 37], and many other domains [17, 20, 36]. Accordingly, graph-based multi-view semi-supervised classification augments the model's potential by leveraging the similarity relationships within data. Among various graph-based multi-view semi-supervised classification algorithms, the graph convolutional network based approach meets researchers' needs for deep models.

Graph Convolutional Networks (GCNs) have garnered considerable attention for their capacity to model both data topology and feature representation. These capabilities have rendered GCNs crucial in a diverse array of domains., including bioinformatics [10, 14, 40], action recognition [21, 25, 41], and traffic forecasting [1, 13, 16]. Researchers have integrated GCNs with multi-view learning by constructing inter-sample relationships directly from the data and extracting consistent representations across different views, as shown in Figure 1 (a). Despite these promising advancements, these methods still face the following challenges: Building a static data dependency within individual view restricts interactions to samples within each view, overlooking potential cross-view information exchange. Additionally, there is still a lack of theoretical understanding regarding how constructing data dependencies can guide multi-view representations to achieve consistency.

To address the outlined challenges, we establish a connection between multi-view learning and GCNs through the lens of graph neural diffusion. Initially, we conceptualize multi-view data as a

multivariate heat diffusion system, where heat can freely propagate within each view and between corresponding points in different views. We model this process through a graph diffusion equation. We then propose a system energy function aimed at guiding inter-view and intra-view diffusion towards a consistent direction for multi-view representations, as demonstrated in Figure 1 (b). Therefore, we develop a framework called Energy Constrained Multi-view Graph Diffusion (ECMGD). Specifically, the proposed method extend graph diffusion to multi-view scenarios by introducing feature flow across views. We then numerically discretize the modeled diffusion equation to derive an updated formula representing each view. To achieve global consistency, we introduce a energy function for multi-view data as a regularization tool. This function ensures that diffusion paths maintain consistent feature propagation both within and across views. Through rigorous mathematical analysis, we demonstrate the intrinsic equivalence between the discretized form of the proposed multi-view diffusion equation and the dynamic minimization of the energy function. A defining characteristic of the proposed framework is its dual diffusion function, which is guided by the system energy. This includes an intra-view diffusion function that facilitates feature propagation within the same view for dynamic instance interactions, as well as an inter-view diffusion function specifically designed to enable precise feature transfer across different views of a particular instance. The contribution of this paper can be summarized as:

- Propose the ECMGD framework to effectively address the deficiency in inter-view perception when constructing data dependencies in multi-view data.
- Provide a multi-view energy function to guide the representation update, and it is mathematically demonstrated that ECMGD enables movement toward consistency.
- Experimental results demonstrate that the proposed model achieves promising results in comparison to state-of-the-art baselines on several datasets.

## 2 RELATED WORK

This section briefly reviews the topics related to this work, including graph-based multi-view learning and graph diffusion models.

### 2.1 Graph-based Multi-view Learning

Graph-based multi-view learning has emerged as a widely adopted learning paradigm. Its fundamental objective is to effectively propagate labels across different views of data by leveraging carefully constructed sample similarity matrics. Satchidanand et al. [28] introduced an approach by employing the extended uncertain random walk framework to facilitate reasoning about multi-relational data. Hao et al. [32] proposed to enhance the learning process of individual view graph matrices and unified graph matrices, ultimately leading to the development of a multi-view fusion technique. Fan et al. [6] proposed a task-directed One2Multi graph autoencoder clustering framework that effectively reconstructs multiple graph views by learning node embeddings using one infographic view and content data. Liang et al. [19] introduced a min-max formulation for graph-based multi-view clustering. Subsequently, they transformed this formulation into a convex and differentiable objective function, enabling the utilization of a simplified gradient descent

algorithm to efficiently reach the global optimum. Huang et al. [11] introduced an attention allocation method to enhance the efficacy of graph-based multi-view clustering, utilizing both node attribute similarity and self-supervised information to comprehensively assess node relevance. These methods demonstrate that graph-based multi-view learning yields superior results compared to traditional approaches.

### 2.2 Graph Neural Diffusion

Graph neural diffusion refers to a diffusion process that is guided by partial differential equations (PDEs). Eliasof et al. [5] drew inspiration from the numerical solution method of PDEs on manifolds to propose PDE-GCN which aimed at mitigating the oversmoothing phenomenon observed in graph convolutional networks. Chamberlain et al. [2] established a connection between layer structure and topology with discretized choices of time and space operators, addressing multiple challenges in graph learning including depth, oversmoothing, noise perturbations, and bottlenecks. Zhao et al. [42] put forward an approach to automatically learn the optimal neighborhood size from the data, challenging the traditional assumption that all GNN layers and feature channels should be propagated using the same neighborhood size. Song et al. [29] extensively investigated the use of thermal semigroups to delve into the enhanced robustness of graph neural PDEs against topological perturbations and introduced a generalized graph neural PDE framework to define a class of robust GNNs. Thorpe et al. [30] introduced graph neural diffusion with source terms, a novel method for deep learning on graphs with a limited number of labeled nodes and without oversmoothing. Huang et al. [9] pioneered the development of a comprehensive node diffusion model known as NDM which is adept at capturing the distinct attributes of individual nodes within the diffusion process, consequently facilitating the creation of top-notch node representations. Although graph neural diffusion models have garnered considerable success in various domains, their extension to complex multi-view learning fields remains challenging due to the intricate coupling among views that cannot be overlooked.

## 3 METHOD

In this section, we elaborate on the proposed methodology, beginning with the graph diffusion process and expanding it to encompass multi-view graph diffusion. We propose an energy function for the multi-view data to guide the representations of views towards consistency. Figure 2 demonstrates the proposed model in detail.

### 3.1 Revisiting Graph Diffusion Process

Let $\mathcal{G} = (\mathbf{A}, \mathbf{X})$ represents the graph, where $\mathbf{A} \in \mathbb{R}^{N \times N}$ and $\mathbf{X} = [x_1; \cdots ; x_N] \in \mathbb{R}^{N \times D}$. Here, $N$ corresponds to the number of samples and $D$ denotes the number of dimensions. Drawing inspiration from thermal diffusion on Riemannian manifolds, all instances are treated as a cohesive entity and propagated as a continuous flow of features. The smoothness of feature propagation between two instances is directly proportional to the disparity in their respective feature sets. Mathematically, this diffusion process can be formally

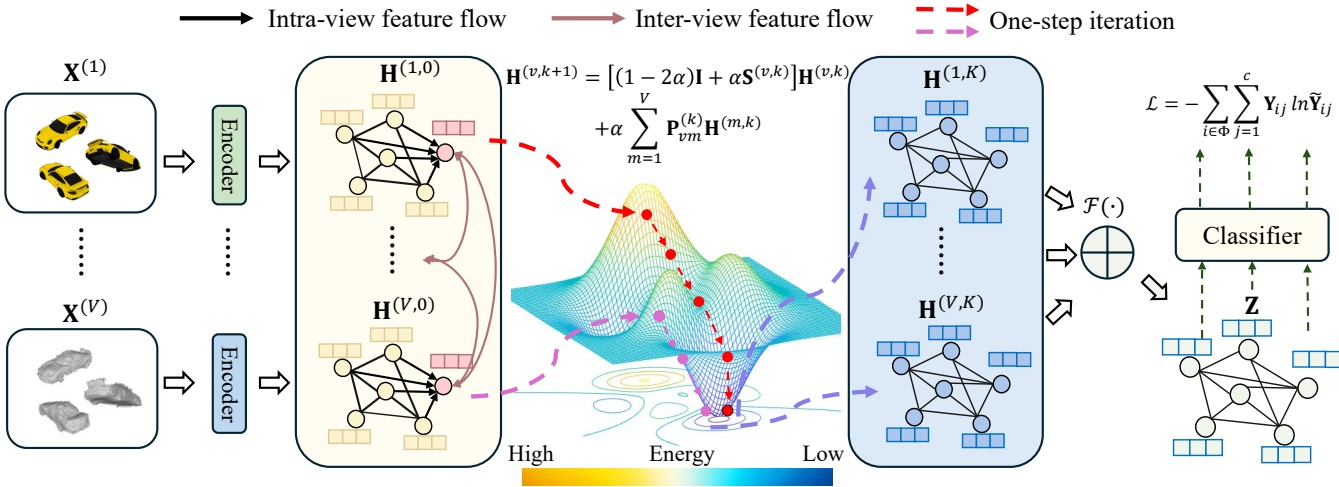

**Figure 2: The proposed ECMGD framework facilitates graph embeddings towards lower energy directions by orchestrating intra- and inter-view diffusion.**

described as follows:

$$
\begin{cases}
\dfrac{\partial \mathbf{H}(t)}{\partial t} = div(\mathbf{A} \odot \nabla \mathbf{H}(t)) \\
\mathbf{H}(0) = \mathbf{X},
\end{cases}
\tag{1}
$$

where $\mathbf{H}(t)$ represents the feature representation at time $t$. The symbol $\odot$ denotes Hadamard product, $\mathbf{A}_{ij}$ denotes the smoothness of feature propagation between instances $i$ and $j$, $\nabla$ indicates the difference between instances, and $div(\cdot)$ represents the cumulative feature flow. More specifically, for the $i$-th instance, the heat flow per unit of time into its interior corresponds to the summation of the heat changes over its space. Equation (1) can be written explicitly as

$$
\frac{\partial \mathbf{h}_i(t)}{\partial t} = \sum_{j=1}^{N} \mathbf{A}_{ij}(\mathbf{h}_j(t) - \mathbf{h}_i(t)).
\tag{2}
$$

Since Equation (2) represents continuous dynamics, practical implementation requires the utilization of numerical methods for its solution. We extend this process to encompass more intricate multi-view scenarios.

### 3.2 The Proposed Approach

Denote multi-view data as $\{\mathbf{X}^{(v)} \in \mathbb{R}^{N \times D^v}\}_{v=1}^{V}$, where $D^v$ is the dimensions of the $v$-th view. Denote heterogeneous graph data $\mathcal{G} = \{\mathbf{A}^{(1)}, \cdots, \mathbf{A}^{(V)}, \mathbf{X}\}$, where $\mathbf{A}^{(v)} \in \mathbb{R}^{N \times N}$ represents the $v$-th meta-path. Let $\mathbf{Y} \in \mathbb{R}^{N \times c}$ be the label matrix and $c$ denotes the number of classes. $\| \cdot \|_2$ denotes the Euclidean norm of vector and $\| \cdot \|_F$ represents the Frobenius norm of matrix. The proposed model begins with a diffusion process that considers the multi-view dataset as a unified entity, facilitating diffusion within and among views via the flow of features. Given that each view in the multi-view data exhibits distinct dimensions, potentially impeding feature flow, we address this by independently mapping the features of

each view into a shared space. The specifics are outlined below:

$$
\mathbf{H}^{(v)} = \mathbf{X}^{(v)} \mathbf{W}^{(v)} + \mathbf{b}^{(v)},
\tag{3}
$$

where $\mathbf{W}^{(v)} \in \mathbb{R}^{D^v \times d}$ and $\mathbf{b}^{(v)} \in \mathbb{R}^{d}$ are trainable weight matrices and bias. In heterogeneous graph data, there is no need for feature dimension alignment. Instead, information from multiple meta-paths is fused into features to create the multi-view format. We can substitute formula (3) with the subsequent equation:

$$
\mathbf{H}^{(v)} = \text{GNN}(\mathbf{A}^{(v)}, \mathbf{X}),
\tag{4}
$$

where $\text{GNN}(\cdot)$ can be a simple GNN architecture, such as GCN. After undergoing the aforementioned process, we expand Equation (2) into the multi-view formulation. The details can be expressed as follows:

$$
\frac{\partial \mathbf{H}_i^{(v)}(t)}{\partial t} = \sum_{j=1}^{N} \mathbf{S}_{ij}^{(v)}(t) \left( \mathbf{H}_j^{(v)}(t) - \mathbf{H}_i^{(v)}(t) \right) +
$$
$$
\sum_{m=1}^{V} \mathbf{P}_{vm}(t) \left( \mathbf{H}_i^{(m)}(t) - \mathbf{H}_i^{(v)}(t) \right),
\tag{5}
$$

where $\mathbf{S}_{ij}^{(v)}(t)$ denotes the diffusion flow coefficient between positions $i$ and $j$ within the $v$-th view at time $t$, while $\mathbf{P}_{vm}(t)$ denotes the diffusion flow coefficient between the $v$-th view and the $m$-th view at position $i$ at time $t$. The interpretation of Equation (5) is as follows: it represents the rate of change of feature at position $i$ for $v$-th view. This change is determined by the summation of feature fluxes entering position $i$ from other positions within the same view in space, as well as the feature fluxes from position $i$ of other views entering position $i$ of the given view. We rely on a variety of ODE solvers for solving differential equations, including the explicit Euler method, the modified Euler method, and the 4-th order Runge-Kutta (RK) method. For example, the explicit Euler

method with step size $\alpha$ is as follows:

$$
\mathbf{H}^{(v,k+1)} = \underbrace{\left[(1-2\alpha)\mathbf{I} + \alpha\mathbf{S}^{(v,k)}\right]\mathbf{H}^{(v,k)}}_{self-updating} + \underbrace{\alpha\sum_{m=1}^{V}\mathbf{P}_{vm}^{(k)}\mathbf{H}^{(m,k)}}_{coupling}. \tag{6}
$$

s.t. $\mathbf{S}^{(v,k)}\mathbf{1} = \mathbf{1},\ \mathbf{P}^{(k)}\mathbf{1} = \mathbf{1},\ \mathbf{P}^{(k)} = \mathbf{P}^{(k)^{\top}}.$

THEOREM 3.1. *Equation (6) converges iteratively when* $0 < \alpha < 1$.

PROOF. The updating rules defined in Equation (6) can be rewritten as

$$
\mathbf{H}^{(v,k+1)} = \mathbf{Q}^{(v,k)}\mathbf{H}^{(v,k)} + \alpha\sum_{m=1}^{V}\mathbf{P}_{vm}^{(k)}\mathbf{H}^{(m,k)}, \tag{7}
$$

where $\mathbf{Q}^{(v,k)}$ and $\mathbf{P}^{(k)}$ are the intra-view and inter-view diffusion matrices, respectively. To ensure convergence, the spectral radius $\rho$ of the update matrix must satisfy $\rho(\cdot) < 1$. For the term involving $\mathbf{Q}^{(v,k)}$, with $\lambda_{\max}(\mathbf{S}^{(v,k)}) = 1$, convergence requires:

$$
|(1-2\alpha) + \alpha\lambda_{\max}| < 1 \Rightarrow 0 < \alpha < 1. \tag{8}
$$

This ensures a contraction in the vector space. Given $\mathbf{P}^{(k)}$ is doubly stochastic with $\lambda_{\max}(\mathbf{P}^{(k)}) = 1$, it maintains the norm of $\mathbf{H}^{(v,k)}$, thus reinforcing the constraint on $\alpha$ to keep $\rho$ of the overall update matrix below 1. □

Given the distinct nature of multi-view data in comparison to graph data, it encompasses a broader spectrum of heterogeneous information yet does not inherently possess a topological structure. Hence, the development of intra- and inter-view diffusion coefficients is critical for facilitating a coherent diffusion process. We define a energy of the multi-view diffusion system as follows:

$$
E(\{\mathbf{H}^{(v)}\}_{v=1}^{V}) = \underbrace{\frac{1}{2}\sum_{v=1}^{V}\sum_{i,j}^{N}\eta(\|\mathbf{H}_{i}^{(v)} - \mathbf{H}_{j}^{(v)}\|_{2}^{2})}_{intra-view} + \\
\underbrace{\frac{1}{4}\sum_{m,n}^{V}\delta(\|\mathbf{H}^{(m)} - \mathbf{H}^{(n)}\|_{F}^{2})}_{inter-view}, \tag{9}
$$

where $\eta(\cdot)$ and $\delta(\cdot)$ denote the monotonically increasing concave functions. The first term quantifies the variance among nodes within the same view, where a minimal variance correlates with reduced energy levels. Similarly, the second term quantifies the disparity between nodes across different views, with the principle that a lesser disparity also results in decreased energy. This dual-term representation underlines the system's equilibrium, emphasizing the importance of minimizing both intra- and inter-view discrepancies to achieve optimal energy efficiency.

$$
\mathbf{H}^{(v,k+1)} = \left[(1-2\alpha)\mathbf{I} + \alpha\mathbf{S}^{(v,k)}\right]\mathbf{H}^{(v,k)} + \alpha\sum_{m=1}^{V}\mathbf{P}_{vm}^{(k)}\mathbf{H}^{(m,k)}.
$$

s.t. $\mathbf{S}^{(v,k)}\mathbf{1} = \mathbf{1},\ \mathbf{P}^{(k)}\mathbf{1} = \mathbf{1},\ \mathbf{P}^{(k)} = \mathbf{P}^{(k)^{\top}},$

$$
E(\{\mathbf{H}^{(v,k)}\}_{v=1}^{V}) < E(\{\mathbf{H}^{(v,k+1)}\}_{v=1}^{V}). \tag{10}
$$

Addressing the equation presented necessitates navigating through an extensive array of constraints, posing significant challenges in

deriving appropriate values for $\mathbf{S}^{(v,k)}$ and $\mathbf{P}^{(k)}$. According to [35] we can learn that the proposed energy upper bound is:

$$
\tilde{E}(\{\mathbf{H}^{(v)}\}_{v=1}^{V}) = \frac{1}{2}\sum_{v=1}^{V}\sum_{i,j}^{N}\left[\mathbf{S}_{ij}^{(v)}\|\mathbf{H}_{i}^{(v)} - \mathbf{H}_{j}^{(v)}\|_{2}^{2} - \tilde{\eta}(\mathbf{S}_{ij}^{(v)})\right] \\
+ \frac{1}{4}\sum_{m,n}^{V}\left[\mathbf{P}_{mn}\|\mathbf{H}^{(m)} - \mathbf{H}^{(n)}\|_{F}^{2} - \tilde{\delta}(\mathbf{P}_{mn})\right], \tag{11}
$$

where $\tilde{\eta}(\cdot)$ and $\tilde{\delta}(\cdot)$ correspond to the conjugate functions of $\eta(\cdot)$ and $\delta(\cdot)$. The upper bound is realized if and only if the conditions are satisfied:

$$
\tilde{\mathbf{S}}_{ij}^{(v)} = \frac{\partial\eta(\mathbf{L}^{2})}{\partial\mathbf{L}^{2}}\bigg|_{\mathbf{L}^{2}=\|\mathbf{H}_{i}^{(v)}-\mathbf{H}_{j}^{(v)}\|_{2}^{2}}, \tilde{\mathbf{P}}_{mn} = \frac{\partial\delta(\mathbf{G}^{2})}{\partial\mathbf{G}^{2}}\bigg|_{\mathbf{G}^{2}=\|\mathbf{H}^{(m)}-\mathbf{H}^{(n)}\|_{F}^{2}}, \tag{12}
$$

where $\mathbf{S}_{ij}^{(v)} = \frac{\tilde{\mathbf{S}}_{ij}^{(v)}}{\sum_{j=1}^{N}\tilde{\mathbf{S}}_{ij}^{(v)}}$. In this paper, we specify the function $\eta(x) = \delta(x) = x - 2\log(e^{\frac{x}{2}-1} + 1)$, and then Equation (12) can be rewrite as:

$$
\tilde{\mathbf{S}}_{ij}^{(v)} = \frac{1}{1 + e^{-f(\mathbf{H}_{i}^{(v)},\mathbf{H}_{j}^{(v)})}}, \tilde{\mathbf{P}}_{mv} = \frac{1}{1 + e^{-g(\mathbf{H}^{(m)},\mathbf{H}^{(v)})}}, \tag{13}
$$

where $f : \mathbb{R}^{d}\times\mathbb{R}^{d} \to \mathbb{R}$, $g : \mathbb{R}^{N\times d}\times\mathbb{R}^{N\times d} \to \mathbb{R}$. To ensure the inter-view diffusion matrix $\mathbf{P}$ retains symmetry and bi-randomness throughout the computation, we employ the differentiable projection algorithm as proposed by Chen et al. (2023) [4]:

$$
\mathcal{J}_{0}(\tilde{\mathbf{P}}) = \frac{\text{softmax}_{dim=0}(\tilde{\mathbf{P}}) + \text{softmax}_{dim=1}(\hat{\mathbf{P}})}{2}, \tag{14}
$$

$$
\mathcal{J}_{1}(\tilde{\mathbf{P}}) = \text{ReLU}(\tilde{\mathbf{P}}), \tag{15}
$$

$$
\mathcal{J}_{2}(\tilde{\mathbf{P}}) = \tilde{\mathbf{P}} - \frac{1}{V}(\tilde{\mathbf{P}}\mathbf{1} - \mathbf{1})\mathbf{1}^{\top}, \tag{16}
$$

$$
\mathcal{J}_{3}(\tilde{\mathbf{P}}) = \hat{\mathbf{P}} - \frac{1}{V}\mathbf{1}(\mathbf{1}^{\top}\tilde{\mathbf{P}} - \mathbf{1}^{\top}), \tag{17}
$$

where $\mathbf{1}$ denotes all 1 vector. Equation (15) - (17) undergoes iterative computations, necessitating a large number of iterations to satisfy the desired conditions. To expedite the convergence of this process, Equation (14) utilizes an initialization to approximate the constraint-satisfying matrix $\tilde{\mathbf{P}}$. Ultimately, we obtain $\mathbf{P} = \mathcal{J}_{3}(\mathcal{J}_{2}(\mathcal{J}_{1}(\mathcal{J}_{0}(\tilde{\mathbf{P}}))))$. By selecting both intra-view and inter-view diffusion functions, the energy function reaches its upper bound. We compute the energy function $E(\{\mathbf{H}^{(v)}\}_{v=1}^{V})$ partial derivative with respect to $\mathbf{H}^{(v)}$. Following this computation, we implement a gradient descent algorithm in a step-wise manner, adopting a step size denoted by $\gamma$, as delineated below:

$$
\mathbf{H}^{(i,k+1)} = \mathbf{H}^{(i,k)} - \gamma\frac{\partial E(\{\mathbf{H}^{(v,k)}\}_{v=1}^{V})}{\partial\mathbf{H}^{(i,k)}} \\
= \left[(1-2\gamma)\mathbf{I} + \gamma\mathbf{S}^{(i,k)}\right]\mathbf{H}^{(i,k)} + \gamma\sum_{m=1}^{V}\mathbf{P}_{im}^{(k)}\mathbf{H}^{(m,k)}. \tag{18}
$$

The detailed calculation process of Equation (18) is shown in **Appendix A**. Equation (18) reveals its structural similarity to Equation (6), indicating that executing a single iteration of the update process effectively corresponds to a reduction in the overall system energy.

After performing $K$ iterations, we attain the final potential representation for each view. Subsequently, we fuse the representation from all views, culminating in the final representation denoted as:

$$Z = \mathcal{F}(\mathbf{H}^{(1,K)}, \cdots, \mathbf{H}^{(V,K)}), \tag{19}$$

where $\mathcal{F}(\cdot)$ is a fuse function, typically implemented as sum, average, and concatenation. We ultimately employ a Multilayer Perceptron (MLP) to map the fused representation onto the probabilities of each category, as outlined below:

$$\tilde{Y} = \text{MLP}(Z), \tag{20}$$

where MLP is parameterized with $\mathbf{W} \in \mathbb{R}^{d \times c}$ and $\mathbf{b} \in \mathbb{R}^c$.

For a semi-supervised classification task, the proposed method employs a loss function defined by the cross-entropy errors:

$$\mathcal{L} = -\sum_{i \in \Phi} \sum_{j=1}^{c} Y_{ij} ln \tilde{Y}_{ij}, \tag{21}$$

where $\Phi$ is the set of samples with labels. The procedural steps of the proposed method can be summarized in Algorithm 1.

---

**Algorithm 1** Energy-Constrained Multi-view Graph Diffusion

---

**Input:** Multi-view data $\mathcal{X} = \{\mathbf{X}^{(1)}, \cdots, \mathbf{X}^{(V)}\}$, label set $\mathbf{Y}$, the hyperparameters $K$ and $\alpha$.

**Output:** Predictive output $\tilde{Y}$.

1: Initialize $\{\mathbf{W}^{(v)}, \mathbf{b}^{(v)}\}_{v=1}^{V}$ and $\{\mathbf{W}, \mathbf{b}\}$ of the networks;
2: **while** not convergent **do**
3:     **for** $v = 0 \rightarrow V$ **do**
4:         Compute $\mathbf{H}^{(v,0)}$ by Equation (3)
5:         **for** $k = 1 \rightarrow K$ **do**
6:             Compute $\mathbf{S}^{(v,k)}$ and $\mathbf{P}^{(k)}$ by Equation (13);
7:             Re-normalize $\mathbf{P}^{(k)}$ by Equation (14) - (17)
8:             Compute $\mathbf{H}^{(v,k)}$ by Equation (10);
9:         **end for**
10:     **end for**
11:     Compute $Z$ by Equation (19);
12:     Compute $\tilde{Y}$ by Equation (20);
13:     Compute $\mathcal{L}$ by Equation (21);
14:     Optimize $\{\mathbf{W}^{(v)}, \mathbf{b}^{(v)}\}_{v=1}^{V}$ and $\{\mathbf{W}, \mathbf{b}\}$ of the networks with backward propagation;
15: **end while**
16: **return** Predictive output $\tilde{Y}$.

---

## 4 EXPERIMENT

In this section, we evaluate ECMGD for two tasks: 1) multi-view data comprising multiple observable modal feature matrices, and 2) heterogeneous graph data containing multiple heterogeneous graphs and one observable feature matrix. For each task, we benchmark ECMGD against a range of closely related competing models.

### 4.1 Datasets

We evaluate ECMGD on 8 real-world multi-view datasets and 4 heterogeneous graph datasets. Among them, BDGP, Flickr, and NUSWIDE are vision-language datasets; HW, GRAZ02, Caltech102, OutScene, NoisyMNIST, and Scene15 are digit image datasets; and

YouTube consists of video games data. For the heterogeneous graph datasets, ACM and DBLP are citation networks; IMDB is a movie dataset; and YELP is a subset derived from a merchant review website. Table 1 illustrates a brief summary of these datasets. Additional details of the datasets can be found in **Appendix B**.

**Table 1: A brief description of multi-view data and heterogeneous graph data.**

| Datasets | #Samples | #Properties | #Views | #Classes |
|----------|----------|-------------|--------|----------|
| BDGP | 2,500 | Multi-modal | 2 | 5 |
| Flickr | 12,154 | Multi-modal | 2 | 7 |
| HW | 2,000 | Multi-view | 6 | 10 |
| GRAZ02 | 1,476 | Multi-view | 6 | 4 |
| Scene15 | 4,485 | Multi-view | 3 | 15 |
| OutScene | 2,688 | Multi-view | 4 | 8 |
| Caltech102 | 9,144 | Multi-view | 6 | 102 |
| Youtube | 2,000 | Multi-view | 6 | 10 |
| ACM | 3,025 | Heterogeneous graph | 3 | 3 |
| DBLP | 4,057 | Heterogeneous graph | 4 | 4 |
| IMDB | 4,780 | Heterogeneous graph | 4 | 3 |
| YELP | 2,614 | Heterogeneous graph | 4 | 3 |

### 4.2 Compared Methods

For multi-view classification, HLR-M$^2$VS [38] and ERL-MVSC [7] are traditional baseline, Co-GCN [18], DSRL [33], LGCN-FF [3], IMVGCN [36], PDMF [12], and GEGCN [23] are networks-based methods. For heterogeneous graph classification, we employ GCN [15], HAN [34], DGI [31], DMGI [26], SSDCM [24], MHGCN [39]. Details of the comparison algorithm can be found in **Appendix C**.

### 4.3 Experimental Settings

All compared methods use the default parameters following the original paper. For the proposed method, we specify the following hyperparameters: The layers number $K = 3$, step size $\alpha = 0.1$, MLP with neuron sizes $[D_v, 64, c]$, the learning rate is set as $5e-3$, training epoch is 200, weight decay set as $5e-5$, and random dropout is 0.5. In multi-view semi-supervised classification, ECMGD leverages a split of 10% supervised samples for the training and 90% unsupervised samples for the testing. Given the absence of a designated validation set, we adopt the strategy of utilizing the model from the final iteration for testing purposes. Conversely, for the semi-supervised classification of heterogeneous graph data, we employ a partitioning scheme of 20% for training, 10% for validation, and 10% for testing. Some experimental results are presented below, with additional results provided in **Appendix D**.

### 4.4 Classification on Multi-view Datasets

**Performance.** The experimental results presented in Table 2 indicate that the proposed algorithm surpasses other algorithms across most datasets. Particularly noteworthy is its superior performance on datasets such as Flickr, Scene15, and Youtube, where it achieves accuracy improvements of 3.1%, 3.5%, and 3.7%, respectively, compared to the algorithm with the second-highest accuracy. The only

**Table 2: Classification results (mean% and standard deviation%) of all compared semi-supervised classification methods with 10% labeled samples as supervision, where the best results are highlighted in red and the second best results are highlighted in blue.**

| Dataset | Metrics | HLR-M$^2$VS | ERL-MVSC | Co-GCN | DSRL | LGCN-FF | IMvGCN | PDMF | GEGCN | ECMGD |
|---|---|---|---|---|---|---|---|---|---|---|
| BDGP | ACC | 94.3 (1.2) | 93.5 (0.8) | 94.6 (1.7) | 98.0 (1.7) | 98.3 (0.2) | 93.3 (0.5) | 90.1 (1.6) | 95.6 (0.7) | 98.1 (0.1) |
| | F1 | 94.3 (1.2) | 93.5 (0.8) | 94.6 (1.7) | 98.0 (1.7) | 98.3 (0.2) | 93.3 (0.5) | 90.1 (1.6) | 95.6 (0.7) | 98.1 (0.1) |
| Flickr | ACC | 56.1 (0.6) | 59.2 (0.5) | 61.2 (2.6) | 67.4 (8.3) | 52.2 (0.5) | 59.1 (0.8) | 64.0 (0.7) | 64.3 (0.1) | 70.5 (0.0) |
| | F1 | 55.6 (0.6) | 59.0 (0.5) | 61.1 (2.4) | 67.2 (8.5) | 52.0 (0.5) | 58.3 (1.0) | 63.2 (0.8) | 64.3 (0.1) | 70.4 (0.0) |
| HW | ACC | 85.3 (0.0) | 87.0 (0.4) | 91.6 (2.7) | 77.9 (0.9) | 92.6 (0.1) | 93.4 (0.8) | 90.0 (2.3) | 94.8 (0.2) | 95.6 (0.4) |
| | F1 | 89.3 (0.3) | 92.7 (0.5) | 86.9 (0.0) | 87.5 (0.3) | 91.5 (2.8) | 93.4 (0.9) | 90.0 (2.3) | 94.8 (0.3) | 95.6 (0.4) |
| GRAZ02 | ACC | 54.7 (2.6) | 54.1 (1.3) | 40.5 (2.6) | 48.1 (1.0) | 49.6 (2.5) | 56.2 (0.5) | 29.7 (1.4) | 61.6 (0.5) | 61.9 (0.2) |
| | F1 | 43.6 (3.6) | 54.8 (1.7) | 56.3 (1.8) | 54.4 (1.2) | 38.9 (1.5) | 48.6 (1.0) | 29.7 (1.4) | 61.5 (0.3) | 61.7 (0.2) |
| Scene15 | ACC | 67.4 (1.3) | 63.1 (1.2) | 58.7 (1.1) | 61.8 (0.9) | 50.1 (4.4) | 65.6 (3.1) | 39.8 (4.6) | 71.8 (0.3) | 75.3 (0.4) |
| | F1 | 67.3 (0.9) | 63.9 (1.3) | 56.7 (0.9) | 60.5 (0.8) | 42.3 (5.7) | 62.0 (2.9) | 39.8 (4.6) | 70.1 (0.3) | 73.8 (0.3) |
| OutScene | ACC | 73.3 (1.3) | 68.8 (1.4) | 71.0 (2.1) | 44.7 (0.8) | 61.1 (11.0) | 77.2 (0.7) | 57.9 (4.7) | 77.6 (0.3) | 79.3 (0.3) |
| | F1 | 75.2 (1.2) | 69.2 (1.4) | 71.3 (2.0) | 42.1 (2.9) | 57.9 (15.6) | 77.4 (0.8) | 57.9 (4.7) | 77.9 (0.3) | 79.3 (0.2) |
| Caltech102 | ACC | 48.1 (0.4) | 50.8 (0.6) | 37.4 (8.7) | 52.9 (0.6) | 40.2 (0.8) | 47.6 (0.1) | 15.3 (0.7) | 51.2 (0.1) | 54.4 (0.3) |
| | F1 | 31.2 (0.7) | 33.8 (0.5) | 20.9 (6.4) | 34.6 (1.2) | 33.4 (0.5) | 24.3 (0.1) | 15.3 (0.7) | 33.7 (0.1) | 35.3 (0.4) |
| Youtube | ACC | 35.9 (6.0) | 45.2 (1.0) | 29.3 (0.3) | 44.7 (0.8) | 47.3 (1.8) | 47.2 (0.6) | 36.9 (3.3) | 55.7 (0.3) | 59.4 (0.4) |
| | F1 | 42.3 (4.0) | 47.9 (0.9) | 21.5 (1.3) | 42.1 (2.9) | 42.3 (5.7) | 45.7 (0.6) | 36.9 (3.3) | 55.7 (0.3) | 59.0 (0.4) |

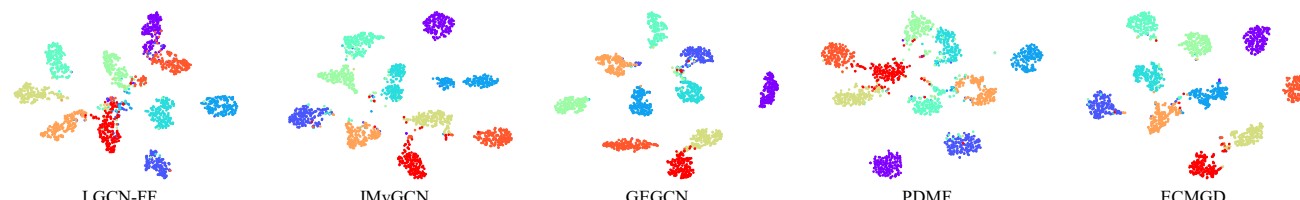

LGCN-FF     IMvGCN     GEGCN     PDMF     ECMGD

**Figure 3: T-sne visualization of LGCN-FF, IMvGCN, GEGCN, PDMF, and ECMGD on dataset HW.**

**Table 3: Classification on large-scale datasets using accuracy as evaluate metric, where 'OOM' denotes Out-of-Memory.**

| Metric | Datasets
Methods / Size | NoisyMNIST
30,000 | NUSWIDE
20,000 |
|---|---|---|---|
| ACC | HLR-M$^2$VS | OOM | OOM |
| | ERL-MVSC | 90.4 (0.0) | 51.2 (0.2) |
| | Co-GCN | 87.9 (1.9) | 63.1 (2.2) |
| | DSRL | OOM | OOM |
| | LGCN-FF | OOM | OOM |
| | IMvGCN | 80.8 (0.1) | 53.2 (1.2) |
| | PDMF | 87.9 (1.2) | 56.8 (0.4) |
| | GEGCN | OOM | OOM |
| | ECMGD | 90.5 (0.1) | 70.8 (0.1) |

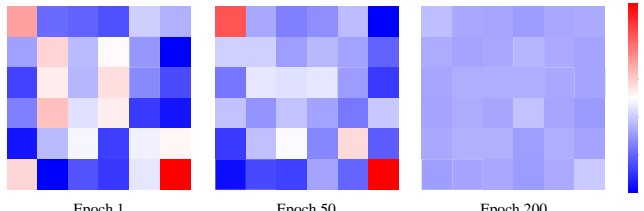

Epoch 1     Epoch 50     Epoch 200

**Figure 4: Visualization of the inter-view diffusion matrix P of ECMGD at various epochs, where red represents the high value and blue denotes the low values.**

**Visualization.** Figure 3 provides a visualization of the classification outcomes across various compared methods applied to the HW dataset. Observation of the figure shows that the proposed method significantly enhances inter-class separability within this dataset. This enhanced separability underscores the superiority of the proposed method in delineating between classes more distinctly. Figure 4 presents a visualization of the inter-view diffusion matrix **P** across varying numbers of layers, offering empirical evidence of the convergence of views with an increasing number of iterations. This

exception is the BDGP dataset, where the proposed algorithm performs slightly lower than LGCN-FF. Furthermore, Table 3 illustrates the proposed method continues to demonstrate competitive performance even on large-scale multi-view datasets.

**Table 4: Accuracy of all compared algorithms on the dataset Youtube under different ratios of supervision, where ↑ represents the gap to the second highest accuracy.**

| Method / Ratio | 0.05 | 0.10 | 0.15 | 0.20 | 0.25 | 0.30 | 0.35 | 0.40 | 0.45 | 0.50 |
|---|---|---|---|---|---|---|---|---|---|---|
| HLR-M$^2$VS | 34.9 | 55.5 | 59.7 | 62.5 | 66.8 | 67.4 | 72.1 | 74.8 | 74.1 | 74.7 |
| ERL-MVSC | 42.4 | 50.7 | 55.7 | 62.3 | 61.7 | 62.0 | 65.0 | 64.8 | 67.9 | 68.7 |
| Co-GCN | 22.2 | 25.6 | 31.9 | 28.6 | 30.9 | 30.5 | 32.7 | 33.8 | 39.0 | 44.7 |
| DSRL | 33.0 | 44.7 | 51.1 | 52.8 | 51.6 | 51.5 | 53.4 | 53.3 | 56.6 | 55.8 |
| LGCN-FF | 40.1 | 40.2 | 47.5 | 45.4 | 48.3 | 49.1 | 52.4 | 55.6 | 52.1 | 56.6 |
| IMvGCN | 48.0 | 56.8 | 62.2 | 61.8 | 66.0 | 66.6 | 65.6 | 65.0 | 67.1 | 67.7 |
| GEGCN | 46.2 | 55.4 | 58.0 | 61.9 | 61.9 | 64.1 | 66.2 | 65.3 | 67.7 | 68.5 |
| PDMF | 31.1 | 33.4 | 38.0 | 33.0 | 39.5 | 36.4 | 37.2 | 37.8 | 40.5 | 48.0 |
| ECMGD | **48.2 (0.2↑)** | **58.4 (1.6↑)** | **67.0 (4.8↑)** | **71.3 (8.8↑)** | **73.3 (6.5↑)** | **76.0 (8.6↑)** | **77.0 (4.9↑)** | **76.3 (1.5↑)** | **77.7 (3.6↑)** | **78.3 (3.6↑)** |

convergence demonstrates the efficacy of the proposed method in harmonizing disparate views toward a consistent representation.

**Traing set size.** Table 4 elucidates the performance accuracy achieved by various algorithms under differing levels of supervision. An observation from this data is that the proposed method not only excels under constrained supervision rates but also significantly surpasses the algorithm with the next highest accuracy by a margin of 8.3% at a 20% supervision rate. This substantial differential underscores the adaptability and efficiency of the proposed algorithm in semi-supervised learning contexts. Such findings compellingly argue in favor of the proposed algorithm's enhanced capability to leverage limited labeled data more effectively.

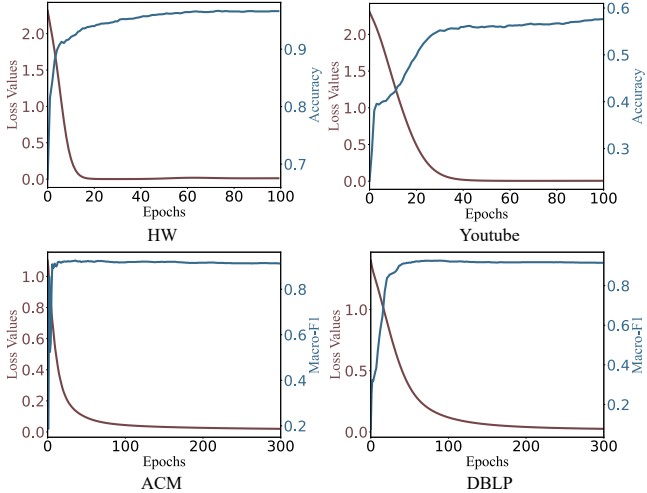

**Figure 5: Loss and accuracy / macro-F1 curves of ECMGD.**

**Convergence.** The convergence behavior of the proposed method is illustrated in Figure 5, which depicts the convergence curves for various datasets. From this figure, there is a gradual decrease in training set loss as the number of epochs increases, alongside a corresponding gradual increase in the accuracy of the test/validation sets. Notably, for the multi-view datasets, satisfactory convergence is achieved at approximately 40 epochs. In contrast, the heterogeneous graph datasets demand a significantly longer duration to reach convergence, typically around 200 epochs.

**Table 5: Classification results of eight compared methods on four benchmark datasets with 20% samples as supervision.**

| Methods | ACM | | DBLP | | IMDB | | YELP | |
|---|---|---|---|---|---|---|---|---|
| | MaF1 | MiF1 | MaF1 | MiF1 | MaF1 | MiF1 | MaF1 | MiF1 |
| GCN | 78.6 | 75.1 | 90.7 | 91.4 | 24.3 | 55.4 | 52.0 | 67.4 |
| SGC | 67.5 | 67.2 | 87.3 | 90.4 | 27.0 | 54.8 | 51.9 | 67.4 |
| DGI | 79.3 | 79.6 | 87.9 | 90.2 | 26.3 | 55.2 | 50.3 | 68.3 |
| HAN | 85.7 | 85.1 | 89.3 | 90.1 | 49.8 | 54.9 | 48.3 | 48.9 |
| DMGI | 87.9 | 87.6 | 90.0 | 90.8 | 35.3 | 57.3 | 51.6 | 69.9 |
| IGNN | 80.9 | 79.5 | 89.1 | 90.2 | 40.3 | 50.0 | 64.2 | 71.2 |
| SSDCM | 87.7 | 87.6 | 89.4 | 89.9 | 49.4 | 59.1 | 52.7 | 70.2 |
| MHGCN | 88.9 | 89.1 | 90.9 | 92.1 | 50.5 | 64.2 | 54.6 | 70.7 |
| ECMGD | 93.0 | 92.9 | 92.3 | 92.7 | 50.0 | 62.6 | 73.9 | 77.8 |

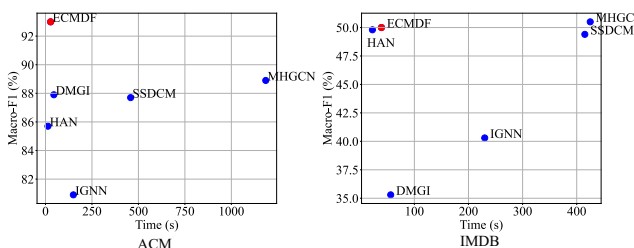

**Figure 6: Running time (seconds) of compared HGNNs with 500 training epochs on dataset ACM and IMDB.**

## 4.5 Classification on Heterogeneous Graph

In this subsection, we extend the experimentation to heterogeneous graph data as shown in Table 5. The results indicate that the proposed method achieves superior performance on datasets ACM, DBLP, and YELP. Notably, the proposed algorithm exhibits slightly lower performance compared to MHGCN on the IMDB dataset.

**Training Time.** In addition, we evaluate the computational efficiency of various algorithms when applied to heterogeneous graph data, with the findings detailed in Figure 6. The comparative analysis reveals that, while the proposed method exhibits a marginal delay in execution time relative to HAN, it nonetheless delivers superior performance outcomes. Significantly, it maintains

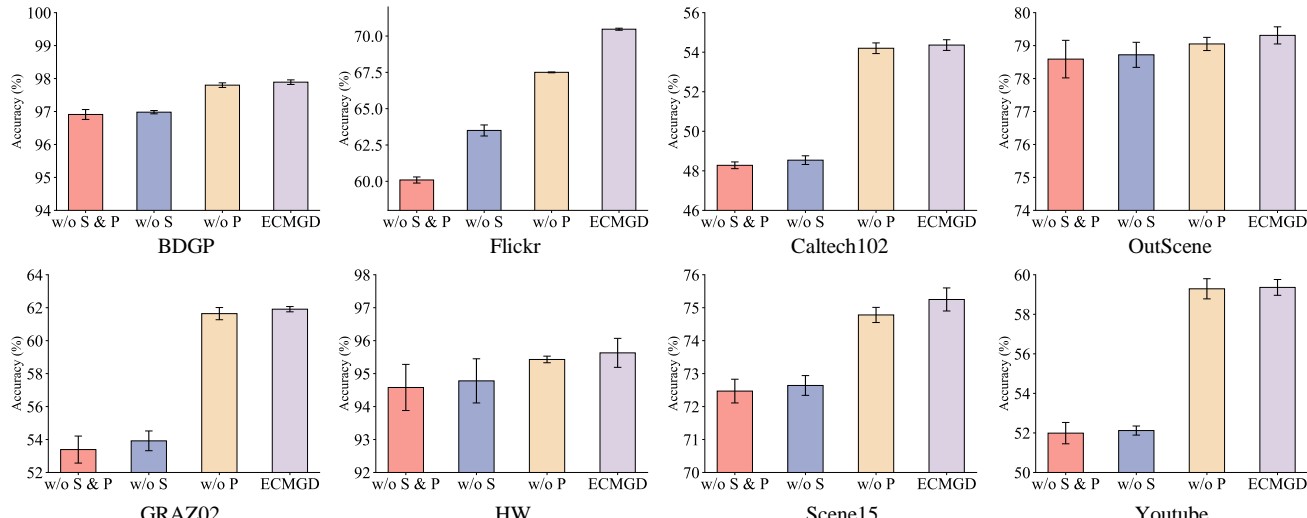

Figure 7: Performance comparisons among model variants on eight datasets.

a considerable speed advantage over both MHGCN and SSDCM. This indicates that the proposed method achieves a clear balance between computational efficiency and performance.

## 4.6 Ablation Study

In this subsection, we evaluate the effectiveness of the proposed method by progressively removing the intra-view diffusion matrix $\mathbf{S}$ and the inter-view diffusion matrix $\mathbf{P}$. For the intra-view diffusion process, we employ the $k$-nearest neighbor ($k$NN) technique to construct the topology of each view. To assess the inter-view diffusion process, we conduct experiments using the identity matrix. The experiment results are shown in Figure 7. Notably, the worst performance is observed when neither $\mathbf{S}$ nor $\mathbf{P}$ is used. Subsequently, the performance of the algorithm improves when either $\mathbf{S}$ or $\mathbf{P}$ is retained, with the highest performance observed when both $\mathbf{S}$ and $\mathbf{P}$ are used. However, an interesting phenomenon emerges: the impact of missing $\mathbf{P}$ on many datasets is much smaller than missing $\mathbf{S}$. However, the absence of $\mathbf{P}$ significantly affects performance on the Flickr dataset. This observation suggests that views in the Flickr dataset may contain more disparate and inconsistent information, necessitating inter-view diffusion to integrate them effectively.

## 4.7 Parameter Sensitivity

In this subsection, we investigate the parameter sensitivity of the proposed model by examining the impact of different step sizes ($\alpha$) and varying numbers of layers ($K$) on the performance of the proposed model. Figure 8 depicts the experimental results. The figure reveals a notable trend: as the value of $\alpha$ increases, the performance of the proposed model decreases while the variance increases. This phenomenon arises from the resemblance of $\alpha$ to the step size utilized in gradient descent. With larger values of $\alpha$, the energy oscillation becomes more pronounced, especially as the number of layers ($K$) grows. Consequently, this oscillation may lead to convergence toward suboptimal energy states, ultimately resulting in diminished performance.

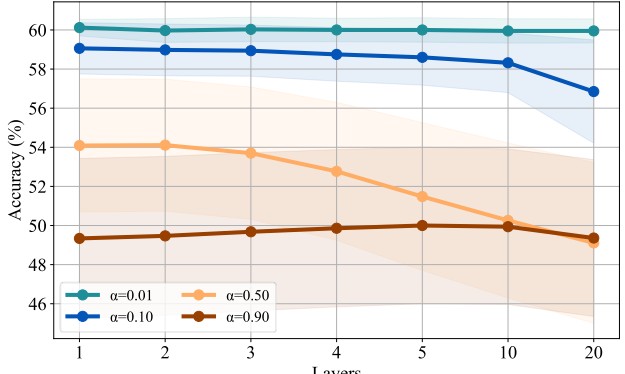

Figure 8: The classification accuracy of ECMGD w.r.t hyper-parameters $\alpha$ and $K$ on Youtube dataset.

## 5 CONCLUSION

In this paper, we introduced a framework called ECMGD, which re-bridges multi-view learning with GCNs through the lens of graph diffusion. This approach effectively closed the gap between GCNs and the pursuit of multi-view consistency, addressing an existing disconnect. Additionally, we proposed an integrated energy function tailored for the multi-view framework as a theoretical basis. The function coordinates intra- and inter-view diffusion to promote cross-view consistency. Through rigorous mathematical derivations, we determined that the proposed iterative algorithm is equivalent to a one-step gradient descent on the energy of the multi-view system. Experimental evaluations on various multi-view datasets and heterogeneous graph datasets confirmed the superiority of ECMGD. Moving forward, we aim to further refine ECMGD to improve its applicability in more complex scenarios.

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
