# OpenReview forum: "Towards Multi-view Consistent Graph Diffusion"
_acmmm.org/ACMMM/2024/Conference — MM2024 Oral_

### Official Review · Reviewer_bTBM · 2024-05-20

**Rating:** 5
**Confidence:** 3

**Summary:**

The authors establish a mathematical correspondence between multi-view data and GCNs via graph diffusion. This approach treats multi-view data as a cohesive entity and devises a feature propagation process that is aware of inter-view interactions, incorporating both inter- and intra-view feature flows across the system. Furthermore, an energy function is introduced to guide this diffusion process, ensuring that the resulting representations achieve global consistency.

**Strengths:**

1). An innovative approach involves treating multi-view data as a cohesive entity and designing a feature propagation process that sensitively integrates interactions between views, encompassing both inter- and intra-view feature flows throughout the system.

2). This is an extremely comprehensive set of baselines to test against, and the performance of ECMGD holds up impressively.

3). The code is concise and clear, and the results can be reproduced.

**Limitations:**

1). Equations 14-17 describe the iterative process; how many iterations are selected for this paper?

2). The meta-path mentioned by the authors in line 281-line 283 requires further explanation.

3). What does the symbol 'S' represent in Section 4.6? If I understand correctly, does it refer to {S^(v)}_{v=1}^V?

4). I am curious about the computational time complexity of the proposed model due to the added diffusion among views.

5). If the value of alpha is made learnable, I believe it could significantly benefit the proposed model.

6). There seems to be some jumps in the derivation of Equation 12- 13. I think it is helpful to detail this process.

**Suitability:**

3

---

### Official Review · Reviewer_JGWH · 2024-05-24

**Rating:** 5
**Confidence:** 3

**Summary:**

To address the limitation of existing methods that only consider static data dependencies within a single view and overlook potential relationships between views, the authors propose a graph diffusion-based multi-view learning approach. Unlike current methods, this approach establishes a mathematical correspondence between multi-view data and Graph Convolutional Networks (GCNs) through graph diffusion. It treats multi-view data as a unified entity and designs a propagation process that is aware of inter-view relationships. Extensive experiments demonstrate the effectiveness of the proposed method.

**Strengths:**

1, The authors propose a graph diffusion method for multi-view learning and conduct extensive theoretical analysis, demonstrating that this method can achieve consistency across different views.
2, The authors conducted extensive quantitative and qualitative experiments on multiple datasets, demonstrating the effectiveness of the proposed method.

**Limitations:**

1, The authors claim that existing methods overlook potential information exchange between views, but the necessity of addressing this issue is not clearly articulated.

2, The authors use a graph diffusion approach to propagate information both within and between views, thereby achieving inter-view information consistency. However, there is a lack of discussion and explanation on whether this process effectively balances the utilization of differential information across the various views.

3, In Equation (9), the energy function of the multi-view diffusion system defined by the authors assigns a weight of 1/2 to the term quantifying intra-view differences, and a weight of 1/4 to the term quantifying inter-view differences. This discrepancy raises the question of whether it results in differing levels of emphasis on intra-view versus inter-view information.

4, Reproducibility is limited. Although the paper presents extensive theoretical proofs, the authors have not provided the corresponding code or implementation details.

**Suitability:**

3

---

### Official Review · Reviewer_ju1K · 2024-06-09

**Rating:** 5
**Confidence:** 3

**Summary:**

In this paper, the authors introduce a novel framework called Energy-Constrained Multi-View Graph Diffusion (ECMGD) to address the limitations of current multi-view learning approaches, particularly their lack of theoretical guidance in constructing multi-view data topologies. By modeling multi-view data as a unified entity and incorporating an energy function to guide both inter-view and intra-view diffusion, the proposed method ensures that representations converge towards global consistency. This framework extends graph diffusion processes to multi-view scenarios, enabling more effective feature propagation across views. Empirical results on various benchmark datasets demonstrate significant performance improvements over state-of-the-art methods.

**Strengths:**

1. This paper is well-written and easy to follow. The authors' keen observations drive their motivation, which is clearly illustrated.
2. The introduction of an energy function to guide the diffusion process is mathematically demonstrated to enhance the consistency of the representations.
3. The paper's effectiveness is validated through extensive experiments on multiple benchmark datasets, showing superior performance compared to existing methods.
4. Thorough ablation studies further support the robustness and superiority of the proposed method over all baselines.

**Limitations:**

1. Section 3.2 is a bit confusing overall; it would be helpful if it could be divided for clarity.
2. The experiments in the paper primarily focus on evaluating semi-supervised classification tasks on several public datasets. However, the method has not been thoroughly assessed for its performance on different data characteristics, such as missing views.

**Suitability:**

2

---

### Meta-Review · Area_Chair_ykN8 · 2024-07-04

**Recommendation:** Accept (Oral)
**Confidence:** 5

**Metareview:**

The paper received three  reviews,  all agreeing  - after the rebuttal phase – on the “accept” recommendation.
Reviewers all state to be confident, and highlight different positive aspects of this submission, thus my recommendation is to accept the paper for oral presentation.